# Single-cell RNA profiling reveals classification and characteristics of mononuclear phagocytes in colorectal cancer

Tiantian Ji[1,2,3☯], Haoyu Fu[1,2,3,4☯], Liping Wang[1,2,3], Jinyun Chen[5], Shaobo Tian[1,2,3,4], Guobin Wang[1,2,3,4]*, Lin Wang[1,2,3,6]*, Zheng Wang[1,2,3,4]*

1 Research Center for Tissue Engineering and Regenerative Medicine, Union Hospital, Tongji Medical College, Huazhong University of Science and Technology, Wuhan, China, 2 Hubei Key Laboratory of Regenerative Medicine and Multi-disciplinary Translational Research, Wuhan, China, 3 Hubei Provincial Engineering Research Center of Clinical Laboratory and Active Health Smart Equipment, Wuhan, China, 4 Department of Gastrointestinal Surgery, Union Hospital, Tongji Medical College, Huazhong University of Science and Technology, Wuhan, China, 5 Department of Transfusion, Union Hospital, Tongji Medical College, Huazhong University of Science and Technology, Wuhan, China, 6 Department of Clinical Laboratory, Union Hospital, Tongji Medical College, Huazhong University of Science and Technology, Wuhan, China

☯ These authors contributed equally to this work.
* wgb@hust.edu.cn (GW); lin_wang@hust.edu.cn (LW); zhengwang@hust.edu.cn (ZW)

**Data Availability Statement:** The single-cell data generated during the current study are available in GEO database under accession code GSE217774.

## Abstract

Colorectal cancer (CRC) is a major cause of cancer mortality and a serious health problem worldwide. Mononuclear phagocytes are the main immune cells in the tumor microenvironment of CRC with remarkable plasticity, and current studies show that macrophages are closely related to tumor progression, invasion and dissemination. To understand the immunological function of mononuclear phagocytes comprehensively and deeply, we use single-cell RNA sequencing and classify mononuclear phagocytes in CRC into 6 different subsets, and characterize the heterogeneity of each subset. We find that tissue inhibitor of metalloproteinases (TIMPs) involved in the differentiation of proinflammatory and anti-inflammatory mononuclear phagocytes. Trajectory of circulating monocytes differentiation into tumor-associated macrophages (TAMs) and the dynamic changes at levels of transcription factor (TF) regulons during differentiation were revealed. We also find that C5 subset, characterized by activation of lipid metabolism, is in the terminal state of differentiation, and that the abundance of C5 subset is negatively correlated with CRC patients' prognosis. Our findings advance the understanding of circulating monocytes' differentiation into macrophages, identify a new subset associated with CRC prognosis, and reveal a set of TF regulons regulating mononuclear phagocytes differentiation, which are expected to be potential therapeutic targets for reversing immunosuppressive tumor microenvironment.

## Author summary

Colorectal cancer (CRC) is a serious health problem worldwide. Mononuclear phagocytes are the main immune cells in the tumor microenvironment of CRC with remarkable

**Funding:** This work was supported by by the National Natural Science Foundation of China (No. 81801578 to TJ, 81974382 to ZW, 82173315 to ZW and 82072743 to GW), the National Key R&D Program of China (No. 2022YFC2408100 to LW) and the Major Scientific and Technological Innovation Projects in Hubei Province (No. 2022BCA013 to ZW). The funders had no role in study design, data collection and analysis, decision to publish or preparation of the manuscript.

**Competing interests:** The authors have declared that no competing interests exist.

plasticity. In this study, single-cell RNA sequencing technology was used to study the subsets and characteristics of mononuclear phagocytes in CRC. We classify mononuclear phagocytes in CRC into 6 different subsets and demonstrate the heterogeneity in infiltrating mononuclear phagocytes of CRC. We find that tissue inhibitor of metalloproteinases (TIMPs) involved in the differentiation of proinflammatory and anti-inflammatory mononuclear phagocytes and reveal the trajectory and characteristics of circulating monocytes differentiation into tumor-associated macrophages (TAMs). We also find a new immunosuppressive subset of mononuclear phagocytes, which is associated with the prognosis of CRC patients. Our findings identify the heterogeneity in infiltrating mononuclear phagocytes of CRC, find a new subset associated with CRC prognosis. Also, we reveal the differentiation of mononuclear phagocytes would lead to an immunosuppressive tumor microenvironment and provide potential targets for reversing the process.

## Introduction

Colorectal cancer (CRC) is one of the most common malignant tumors worldwide [1]. Although much progress has been made in cancer biology and treatment, the prognosis of CRC patients still remains poor. The process of tumor development is not only related to gene mutations and functional changes of tumor cells, but also affected by tumor microenvironment [2]. As a major component of tumor microenvironment, tumor-associated macrophages (TAMs) are remarkably plastic. TAMs reportedly promote tumor progression, invasion, and dissemination through various mechanisms [3].

TAMs, mainly originated from circulating monocytes, are recruited to tumor sites through growth factors and chemokines secreted by tumor cells. In response to various signals in tumor microenvironment, circulating monocytes differentiate into TAMs, and different types of signals within tumor environment (such as a plethora of tumor cell-derived soluble molecules and metabolites) can result in different TAMs gene transcriptional expression profiles and functions. The current classification divides macrophages into classically activated M1 macrophages and alternatively activated M2 macrophages. However, this classification cannot fully cover all functional subtypes of macrophages, and is not suitable for functional classification of TAMs. Some researchers have divided M2 macrophages into M2a, M2b, M2c and M2d subtypes according to external stimulating signals and their main involvement in immune responses [4–8]. Even so, these classifications are insufficient to cover functional heterogeneity of macrophages. To understand immunological functions of TAMs comprehensively and deeply, the study on how to more accurately cluster TAMs was highly needed.

Single-cell RNA sequencing (scRNA-seq) is a powerful new way to investigate transcriptional profiles of individual cells, which has facilitated the exploration of molecular heterogeneity of TAMs. Using single-cell sequencing, some studies found that a higher level of SPP1$^+$ macrophages was correlated with shorter progression-free survival (PFS) in CRC patients [9,10], indicating that scRNA-seq can distinguish different functional subsets in TAMs from gene expression levels at a single-cell resolution. Therefore, the analysis of different subsets of TAMs in CRC using scRNA-seq would deepen our understanding of tumor microenvironment and point to new directions of cancer therapy.

Monocytes and macrophages are two major components of the mononuclear phagocyte system (MPS) [11]. During MPS differentiation, monocytes differentiate into macrophages, and macrophages exhibit different functions according to different physiological or pathological conditions. Study on MPS would reveal dynamic changes from monocytes to pro-tumorigenic TAMs. Here, our study using scRNA-seq revealed MPS' heterogeneity and identified an

immunosuppressive subset of MPS, which was associated with poor prognosis of CRC patients. We also found possible targets for reversing immunosuppressive tumor microenvironment. This work provides new insights into MPS in CRC and might facilitate the development of immunotherapy for CRC patients.

## Results

### Single-cell RNA sequencing reveals heterogeneity of mononuclear phagocytes in CRC

To explore the characteristics of monocyte differentiation into TAM in CRC, we collected the tumor tissues of three CRC patients (WUH1, WUH2 and WUH3), and selected the marker gene CD14 for magnetic bead sorting (MACS). The cells sorted were used to perform scRNA-seq (Fig 1A). After quality control (S1A Fig) and removal of the batch effect (S1B Fig), 5153 cells were identified using SingleR package [12]. All cells were confirmed to be MPS, with marker gene of mononuclear phagocytes (CD14) highly expressed and marker genes of intestinal dendritic cells (CX3CR1) and neutrophils (MPO and CEACAM8, also named CD66b) rarely expressed (S1C Fig) [13–15].Then these cells were clustered into six clusters: cluster 1 (C1) to cluster 6 (C6) (Fig 1B).

Different clusters showed unique gene expression profile (Figs 1C and S2A), reflecting MPS heterogeneity. The existences of six clusters were further validated by immunofluorescence staining in CRC (S2B Fig). C1 was characterized by the high expression level of the S100 family (S100A8, S100A9 and S100A12). Since S100A8 and S100A9 are mainly derived from immune cells and participate in inflammatory process [16], C1 could be involved in the inflammatory process of CRC. C2 had high MALT, SLC25A37, and SLC2A3 expression, and C3 had high PLAC8 expression. C4, C5 and C6 had high expression of complement component C1q (C1QA, C1QB, and C1QC). Given that complement component C1q can stimulate phagocytic function of human monocytes and macrophages [17], these clusters might have stronger phagocytic function. C5 also highly expressed APOE, RNASE1, and APOC1, while C6 highly expressed cell cycle genes (TOP2A, MKI67, UBE2C and CDK1), suggesting that C6 is a proliferative cluster. In addition, MPS also exhibited inter-patient heterogeneity (Fig 1D). WHU1, WHU2, and WHU3 had a higher proportion of C5, C1 and C2, respectively. These results indicate that MPS displays a high molecular heterogeneity in CRC patients.

Functional analysis showed different functional characteristics in each cluster (Fig 1E). C1, C2 and C3 were enriched in pathogen-associated molecular patterns (PAMP) receptor signaling pathway, indicating function of innate immunity [18]. While C4, C5 and C6 showed an enrichment in antigen processing and presentation, indicating their role in adaptive immunity [19]. Besides, C1 was enriched in monocyte aggregation, C5 had ability to clear lipoprotein particle, and C6 showed a stronger cell cycle DNA replication.

### Clustering of mononuclear phagocytes in CRC is different from that in blood and normal colon

In healthy intestinal tissue, MPS consists of blood-derived monocytes and tissue-resident macrophages [20]. To explore the origins of the identified MPS subsets in CRC, we aligned our data with data of blood and normal colon tissue from healthy people and CRC patients. The integrated data showed similar clusters of MPS (Fig 2A, B), while proportion of each cluster varied in different sample and tissue (Fig 2C and 2D). The main MPS cluster in blood was C1 and blood-derived monocyte marker (S100A8) [20] was highly expressed in blood data (Fig 2E), indicating C1 as blood-derived monocytes. In normal colon, C1 and C5 were main MPS

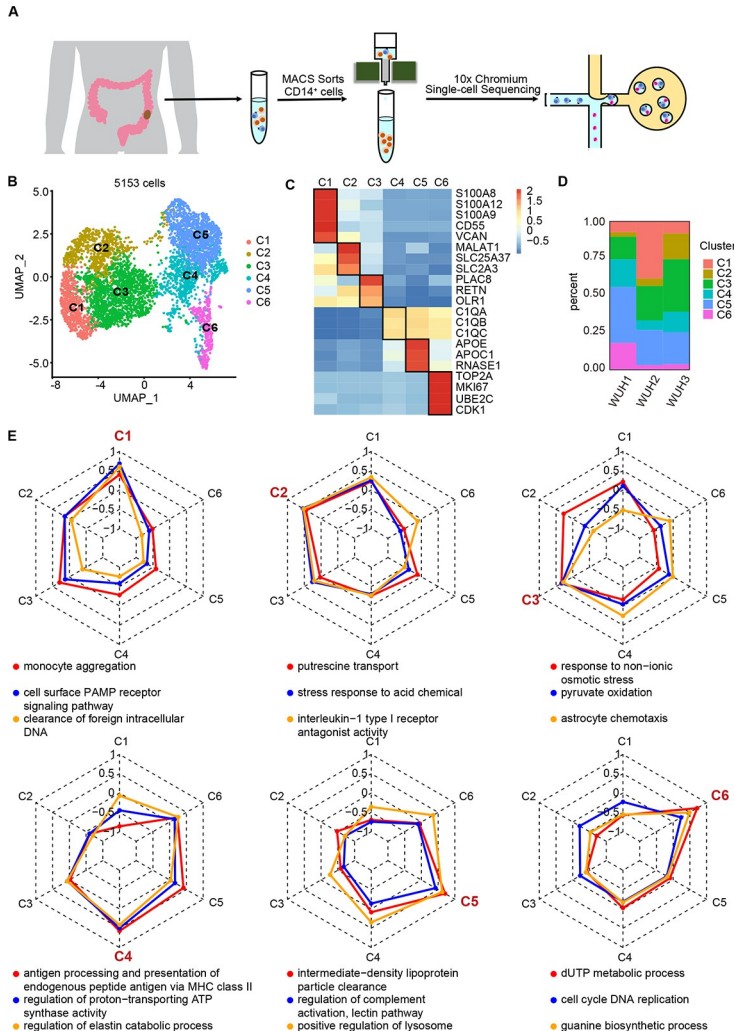

**Fig 1. Identification of mononuclear phagocytes subsets in CRC.** (A). Schematic illustration of the workflow of the sample preparation and scRNA-seq. (B). UMAP plot of mononuclear phagocytes in CRC. Cells are clustered into six clusters and colored by clusters. (C). Heatmap showed the marker genes of six clusters. Gene expression is color-coded according to a scale based on Z score distribution from −0.5 (blue) to 2 (red). (D). Bar plot showed proportion of different clusters in different samples. (E). Radar charts showed GSVA score of biological process GO term in six clusters.

clusters and mucosal macrophages marker (SELENOP) [20] was highly expressed in C5 (Fig 2E), suggesting that C5 was tissue-resident macrophage. Compared with healthy people, normal colon in CRC patient showed a higher proportion of C1. MPS in cancer tissue showed completely different clustering, with proportion of C2 increased and C3, C6 specifically appeared in cancer tissue. These results indicated tumor microenvironment of CRC would result in unique polarization of MPS.

## Tissue inhibitor of metalloproteinases (TIMPs) participate in the differentiation of M1-like and M2-like mononuclear phagocytes

The classic classification groups macrophages into M1 or M2 types. Further studies found, according to different stimuli, the M2 macrophages can be subdivided to M2a, M2b, M2c and

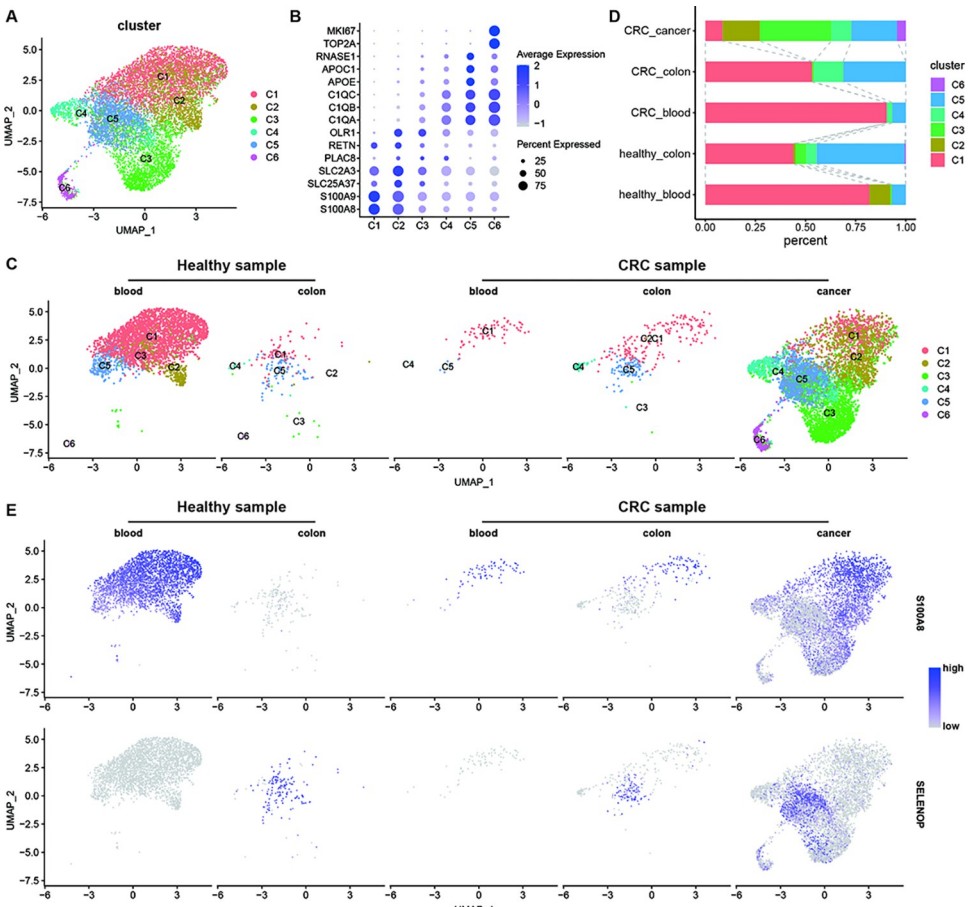

**Fig 2. Comparison of mononuclear phagocytes subsets in blood, normal colon and CRC.** (A). UMAP plot of mononuclear phagocytes from blood, normal colon and CRC from healthy and CRC samples. (B). Dot plot showed marker genes of six clusters. (C and D). Split UMAP plot (C) of mononuclear phagocytes and bar chart (D) showed proportion of six clusters in different sample. (E). Split UMAP plots showed expression levels of markers of intestinal monocytes (S100A8) and mucosal macrophages (SELENOP) in mononuclear phagocytes.

M2d macrophages [21]. However, markers of these subdivisions [6,22,23] could not distinguish MPS in CRC (Fig 3A). To identify more specific markers that could distinguish MPS in human CRC, we calculated the M1 and M2 polarization scores using related gene sets (Fig 3B) and observed M1-like and M2-like cells present in MPS. C1, C2, C3 were M1-like cells, and C4, C5, C6 were M2-like cells (Fig 3C). We also analyzed a public scRNA-seq dataset of CRC (GSE132465) for validation and found the similar results (S3A–S3C Fig).

To explore the characteristics of M1-like and M2-like cells, differential gene expression analysis was performed (S3D Fig). Tissue inhibitor of metalloproteinases (TIMPs) were differentially expressed in M1, M2-like cells. TIMP1 and TIMP2 were highly expressed in M1-like and M2-like cells, respectively (Fig 3D and 3E), which was consistently verified when using the public scRNA-seq dataset (S3E Fig). To further verify this finding, we obtained human blood-derived monocytes and induced them into M1 or M2 macrophages. Q-PCR analysis showed that, compared with M2 macrophages, M1 macrophages expressed higher TIMP1 and lower TIMP2 (S3F Fig).

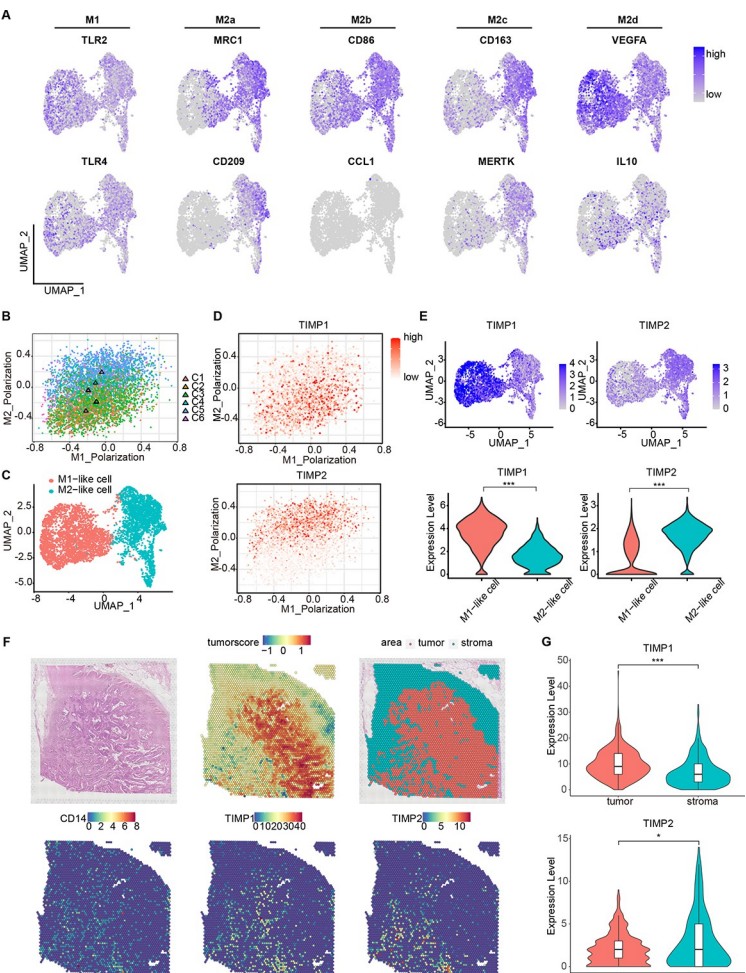

**Fig 3. Differential expression of TIMPs in M1-like cells and M2-like mononuclear phagocytes.** (A). UMAP plots showed expression levels of M1 markers (TLR2, TLR4), M2a markers (MRC1, CD209), M2b markers (CD86, CCL1), M2c markers (CD163, MERTK) and M2d markers (VEGFA, IL10). (B). Scatterplot showed M1 polarization and M2 polarization score of each cluster, the colored triangles coded by clusters present mean score of each cluster. (C). UMAP plot of mononuclear phagocytes in CRC, color-coded by M1-like and M2-like cell. (D). Scatterplots showed expression levels of TIMP1 (top) and TIMP2 (bottom). (E). UMAP plots and violin plots showed expression levels of TIMP1 and TIMP2 in M1-like and M2-like cell. (F). Spatial plots showed areas of tumor and stroma based on tumor score in tissue sections (top), and distribution signature of TIMP1 and TIMP2 in CD14-expressed area (bottom). (G). Violin plots showed expression level of TIMP1 and TIMP2 in different CD14-expressed areas. * represent p value < 0.05, ** represent p value < 0.01, *** represent p value < 0.001.

To decipher the spatial distribution of mononuclear phagocytes with high expression of TIMP1 or TIMP2 in the tumor microenvironment, spatial transcriptome data of CRC was analyzed. Tumor area was identified by tumor score and CD14 was used to label mononuclear phagocytes. As the result shown, MPS with high expression of TIMP1 were distributed more within the tumor, while MPS with high expression of TIMP2 were distributed more in stroma (Figs 3F, 3G and S4A, S4B). Besides, spatial distribution of different clusters varied as well (S4C Fig). Despite being considered as secreted endogenous inhibitors of metalloproteinases, TIMP1 was reported to promote cancer progression [24], but its roles in mononuclear phagocytes remained unclear. Our results not only indicate the complexity of mononuclear phagocytes' classification in human CRC, but also suggest that TIMPs' involvement in mononuclear phagocytes' differentiation.

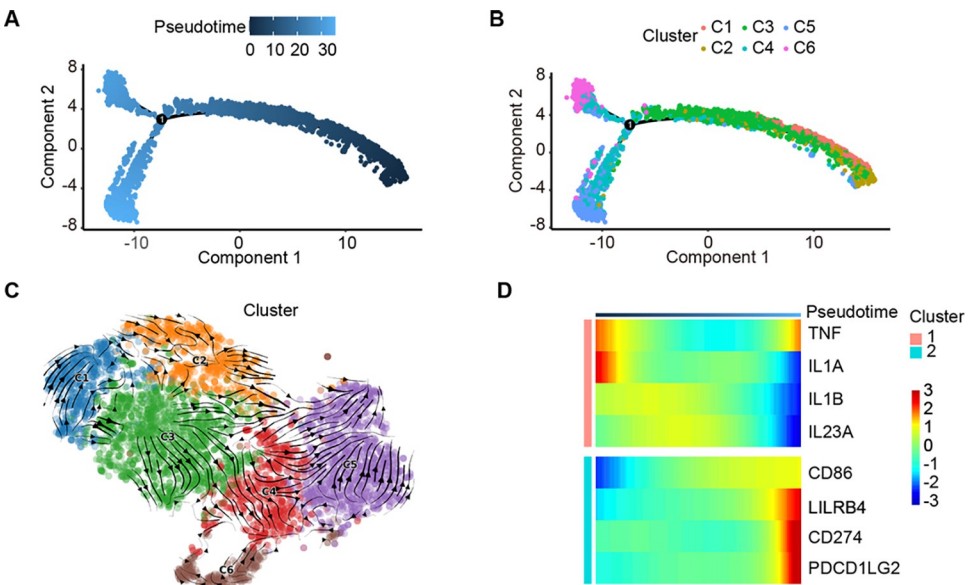

**Fig 4. The trajectory and characteristics of mononuclear phagocytes' differentiation into macrophages.** (A). Pseudotime plot showed the dynamics of mononuclear phagocytes. (B). Pseudotime plot of mononuclear phagocytes showed stages of each cluster. (C). Developmental trajectory of mononuclear phagocytes inferred by RNA velocity. Arrows indicate developmental trend of each cell. (D). Heatmap showed downregulated inflammatory genes and upregulated immunosuppression-related genes in the differentiation of mononuclear phagocytes.

## Trajectory analysis reveals the differentiation characteristics of mononuclear phagocytes into immunosuppressive macrophages

Differentiation of mononuclear phagocytes in CRC is a dynamic process [11]. Pseudotime analysis was performed to investigate this ongoing process (Fig 4A), and showed that different clusters were at different stages of differentiation (Figs 4B and S5A). RNA velocity analysis of mononuclear phagocytes showed that C1 was the initial state of differentiation; C3 and C4 were the transitional clusters; C5 was in the terminal state of differentiation (Fig 4C).

During the differentiation of mononuclear phagocytes, the expression of inflammatory genes (TNF, IL1A, IL1B, IL23A) and immunosuppression-related genes (CD86, LILRB4, CD274 [25] PDCD1LG2) was gradually decreased and increased, respectively (Figs 4D and S5B), indicating that the immune function of mononuclear phagocytes decreases towards forming an immunosuppressive tumor microenvironment. Here we reveal the differentiation characteristics of mononuclear phagocytes that different clusters were at different stages of differentiation and formation of the immunosuppressive tumor microenvironment.

## Changes of transcription factors during of mononuclear phagocytes differentiation

To discover potential targets for inhibiting or reversing the formation of the immunosuppressive tumor microenvironment, we combined SCENIC [26] to analyze the dynamic changes of transcription factors (TFs) and TF regulons (consisting of the TF and their target genes) during differentiation of mononuclear phagocytes. The results showed the relative expression of TFs genes and the normalized activity of TF regulons in each cluster (Fig 5A and 5B). We found that the TF regulons ELF2, RCOR1, CREB1, ETV6, ETV3, CHD1 were downregulated (Fig 5C), and the TF regulons MAF, USF2, NFIC, ETV5, NFATC2, TCF4 were upregulated (Fig 5D).

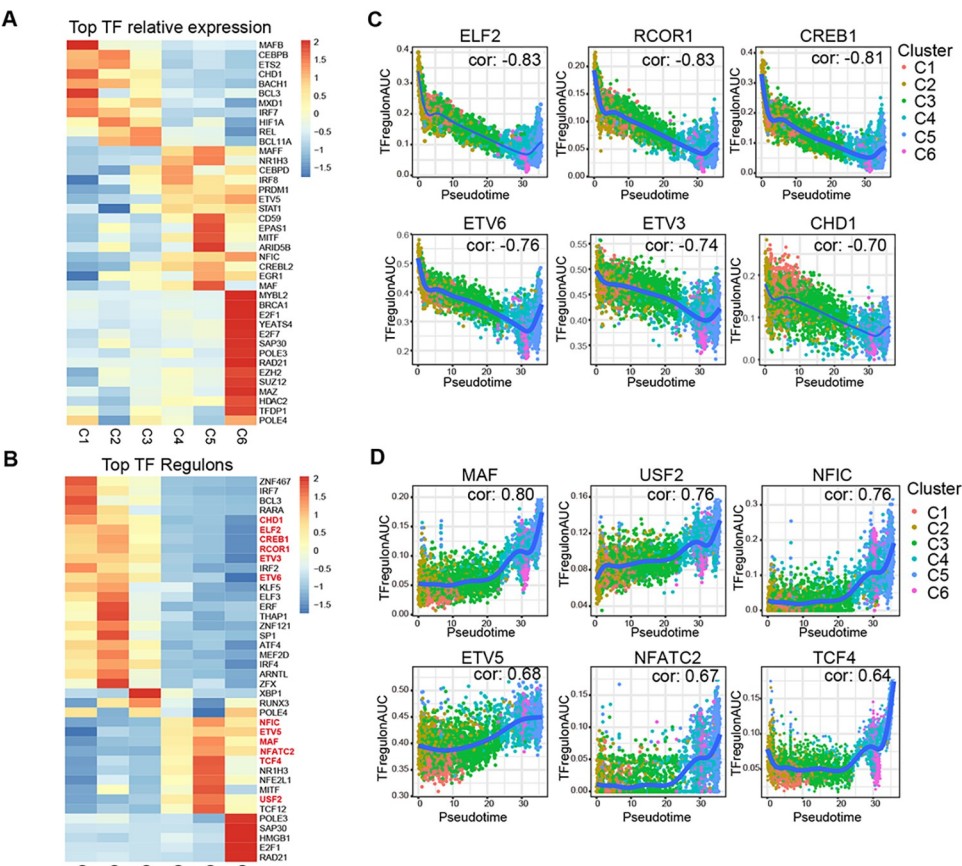

**Fig 5. The Changes of transcription factors during mononuclear phagocytes' differentiation into macrophages.**
(A and B). Heatmaps showed the relative expression (z-score) of top TFs (A) and TF regulons (B) in each cluster. Gene expression level is color-coded according to a scale based on Z score distribution from −1.5 (blue) to 2 (red). (C and D). Scatterplots showed downregulated (C) and upregulated (D) TF regulons in the differentiation of mononuclear phagocytes. Correlation coefficient of pseudotime and AUC score of TF regulons was calculated using the Pearson correlation coefficient presented at top right corner of each scatterplot.

Our results reveal the expression characteristics of TFs and TF regulons during mononuclear phagocytes differentiation, providing potential targets for reversing immunosuppressive tumor microenvironments.

## Terminally differentiated cluster C5 is an immunosuppressive cluster

Trajectory analysis revealed that the differentiation of mononuclear phagocytes might form an immunosuppressive tumor microenvironment and C5 was in the terminal state of the differentiation, suggesting that C5 plays an important role in the formation of an immunosuppressive tumor microenvironment. To explore the role of C5 in tumor treatment, we analyzed publicly available scRNA-seq dataset from CRC patients treated with or without preoperative chemotherapy (PC) (GSE178318) and CRC patients before and after chemotherapy (GSE232525). The public scRNA-seq dataset showed similar clusters of mononuclear phagocytes with our data (S6A–S6D Fig). PC significantly reduced C5 abundance in CRC (Fig 6A and 6B). CRC before and after chemotherapy showed similar results (Fig 6C and 6D). These patients who have undergone treatment showed significant tumor shrinkage, suggesting that C5 is a potential target cluster for CRC treatment.

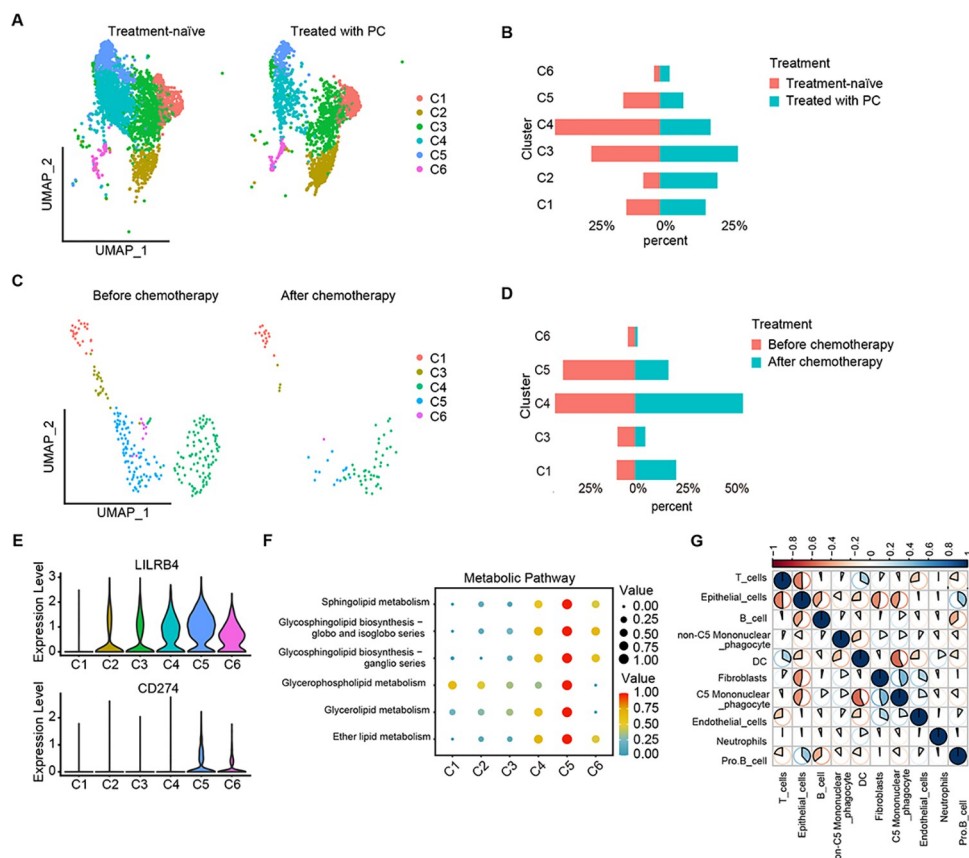

**Fig 6. Unique characteristics of immunosuppressive cluster C5.** (A). UMAP plots of mononuclear phagocytes in CRC with treatment-naïve (left) and treated with PC (right). Cells are colored by clusters. (B). Bar plot showed proportion of different clusters in CRC with treatment-naïve and treated with PC. (C). UMAP plots of mononuclear phagocytes in CRC before chemotherapy (left) and after chemotherapy (right). Cells are colored by clusters. (D). Bar plot showed proportion of different clusters in CRC before and after chemotherapy. (E). Violin plots showed expression levels of LILRB4 (top) and CD274 (bottom) in each cluster. (F). Dot plot showed metabolism activity of different clusters. (G). Pairwise Pearson's correlation for infiltration patterns of ten major cell types in TCGA CRC cohort.

We further explored the characteristics of C5. Consistent with trajectory analysis, our scRNA-seq dataset showed that C5 highly expressed the immunosuppression-related genes (LILRB4, CD274) (Fig 6E), indicating that C5 is an immunosuppressive cluster. ScMetabolism [27] was applied to explore the metabolic characteristics of C5 and found a significant enhancement in lipid metabolism (Fig 6F). In addition, paired Pearson correlation analysis showed that in CRC tumor microenvironment, the fibroblasts and C5 mononuclear phagocytes were the most highly correlated populations in TCGA CRC cohort (Fig 6G).

Some studies also identified SPP[+] macrophages as a pro-tumorigenic/ pro-metastatic cluster in CRC [9,10], but doubts arose as to whether C5 were the same as SPP[+] macrophages. The comparison of gene signature between C5 and SPP[+] macrophages revealed a clear difference between these two cell clusters (S7A and S7B Fig), verifying C5 as a unique cluster. Together, these results indicate that C5 is an immunosuppressive cluster with a significantly activated lipid metabolism.

## C5 is associated with poor prognosis for CRC patients

Immunosuppressive tumor microenvironment can promote cancer progression and affect patient prognosis. To explore the relationship between MPS subsets with CRC prognosis, we

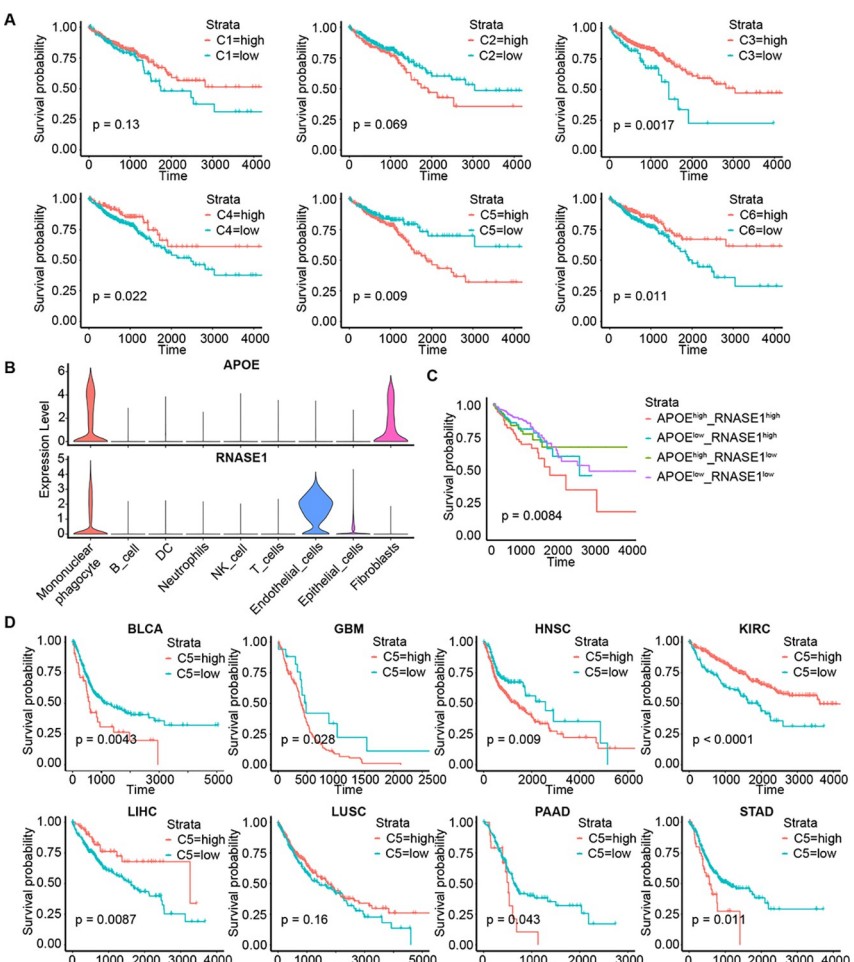

**Fig 7. C5 abundance is negatively correlated with prognosis of CRC patients.** (A). Kaplan–Meier survival curves of TCGA COAD and READ patients grouped by abundance of six mononuclear phagocytes subsets. (B). Violin plots showed expression levels of APOE (top) and RNASE1 (bottom) in different types of cells. (C). Kaplan–Meier survival curves of TCGA COAD and READ patients grouped by the expression levels of APOE and RNASE1. (D). Kaplan–Meier survival curves of TCGA BLCA, GBM, HNSC, KIRC, LIHC, LUSC, PAAD and STAD patients grouped by C5 abundance. P values were calculated using the log-rank test.

correlated our scRNA-seq data to public datasets. Abundance of each cluster in tumor tissue was evaluated with CYBERSORTx, a computational framework for inferring cell type abundance and cell-type-specific transcriptomes from RNA profiles of intact tissue [28]. Results showed only C5 is associated with worse prognosis in CRC (Fig 7A). The public scRNA-seq dataset of CRC showed that APOE and RNASE1, two marker genes of C5, were highly co-expressed in mononuclear phagocytes only (Fig 7B). Survival analyses performed on the TCGA CRC cohort also showed patients with high expression level of APOE and RNASE1 were tightly associated with a worse prognosis (Fig 7B and 7C). These results indicate that C5 is negatively associated with the prognosis of CRC patients.

To expand these observations to more tumors, we explored scRNA-seq datasets of bladder cancer (BLCA), glioblastomas (GBM), head and neck cancer (HNSC), kidney clear cell cancer (KIRC), liver cancer (LIHC), lung cancer (LUSC), pancreatic cancer (PAAD) and gastric cancer (STAD). The results showed that MPS in these tumors had similar clusters with C5 present in all tumors (S8A–S8H Fig). Survival analysis showed C5 was associated with a worse

prognosis of patients with BLCA, GBM, HNSC, PAAD and STAD, with no significance in LUSC patients, and was associated with better prognosis in KIRC and LIHC (Fig 7D).

## Discussion

Many studies have shown that macrophages play an important role in cancer development. CRC scRNA-seq dataset shows that mononuclear phagocytes are highly enriched in cancer tissues. TAMs in CRC are mostly derived from and are maintained by circulating monocytes. A better understanding of how circulating monocytes differentiate into macrophages and the function of various subsets of mononuclear macrophages in CRC may open new avenue for targeting mononuclear macrophages for CRC therapy. Here, scRNA-seq was used to analyze the MPS in CRC. We revealed the trajectory and characteristics during circulating monocytes' differentiation into TAMs, identified the heterogeneity in infiltrating mononuclear phagocytes of CRC, and found a new cluster of mononuclear phagocytes as an immunosuppressive cluster that was associated with CRC patients' prognosis. Also, we revealed the differentiation of mononuclear phagocytes would lead to an immunosuppressive tumor microenvironment and provide potential targets for reversing the process.

Consistent with the studies in other tumors [29], our study found the presence of M1-like and M2-like cells in mononuclear phagocytes of CRC, but the traditional M1 markers and M2 markers were unable to effectively distinguish M1-like cells and M2-like cells. Failure of conventional classification underscored the complexity of tumor microenvironment and required more in-depth research. Differential gene expression analysis of M1-like and M2-like cells showed TIMP1 and TIMP2 was highly expressed in M1-like and M2-like cells, respectively. TIMP1 and TIMP2 are the inhibitors of MMPs. As MMPs were reportedly associated with poor prognosis of cancer patients [30], TIMPs were first thought to be disease-protective molecules. Although TIMP1 can be antitumorigenic and antimetastatic functions in cell lines or mice [31–33], clinical findings showed that TIMP1 was consistently elevated in cancerous tissues as well as in blood, and clearly correlated with poor prognosis of most cancer patients [24,34]. This was consistent with the spatial transcriptome data we analyzed, which showed that MPS with high expression of TIMP1 were distributed more within the tumor, while MPS with high expression of TIMP2 were distributed more in stroma. However, clinical results of TIMP2 varied with tumor types, which might be caused by different interactions and non-anti-proteolytic functions of TIMPs. Current studies on TIMPs are limited to tumor cells, and clinical results of TIMPs do not often take the complex components of TME into account. Thus, the relationship between TIMPs and mononuclear phagocytes remained undetermined. Our study found different expression patterns of TIMP1 and TIMP2 in mononuclear phagocytes, suggesting that the TIMPs may participate in differentiation of mononuclear phagocytes. However, how TIMPs in mononuclear phagocytes affect CRC progression remains unclear and awaits further investigation.

Our trajectory analysis revealed different clusters were at different differentiation stages and would form an immunosuppressive tumor microenvironment during the differentiation of mononuclear phagocytes. To find out potential targets that might reverse the immunosuppressive tumor microenvironment, we analyzed the dynamic changes of TF regulons during the process of differentiation. Several members of ETS family of TF regulons showed dynamic changes. TF regulons ELF2, ETV3, ETV6 were downregulated and ETV5 was upregulated during the differentiation. ETV3 was reportedly a transcriptional repressor that block terminal macrophage differentiation in a Ras-dependent manner [35], while the mechanistic relationship between ETS and mononuclear phagocytes was not clear. Our results suggest the important role of ETS in mononuclear phagocytes, which required further validation. Recently,

therapeutic strategies based on ETS proteins have made much progress. Small molecule inhibitors of ETS proteins exhibited antitumor effects, possibly opening up novel avenues for CRC therapies [36–38]. Among the TF regulons that are downregulated during macrophage differentiation, CREB1 induces the upregulation of anti-proliferative Ets repressor factor [39], while loss of RCOR1 could enhance monocytic cell survival and self-renewal [40]. Consistent with MAF as a critical controller for immunosuppressive macrophage polarization and function in cancer [41,42], MAF was found as a transcription factor regulon upregulated during macrophage differentiation. Together, these TF regulons could be the potential targets that regulate differentiation of mononuclear phagocytes and further reverse the immunosuppressive tumor microenvironment.

In exploring the origins of the identified MPS subsets in CRC, we found that C1 was blood-derived monocyte and mucosal macrophages marker (SELENOP) was highly expressed in C5, suggesting that C5 was tissue-resident macrophage. Immunosuppressive cluster C5, which highly expresses C1QA, APOE and RNASE1, was at the terminal state of differentiation and played an important role in the formation of an immunosuppressive tumor microenvironment [43,44]. With the unique gene signature, C5 was also associated with poor prognosis for CRC patients. C5 had a strong gene enrichment in extracellular matrix disassembly and a significantly activated in lipid metabolism, which could collectively promote angiogenesis and tumor-promoting microenvironment. Thus, blockade of C5 formation could be beneficial in treating CRC. Cortese N, et al. have reported that GPNMB$^+$ TAM is negatively associated with CRC prognosis. GPNMB$^+$ TAM more often accumulated at the tumor margin, therefore more exposed to tumor immunosuppressive signals, favor tumor growth and metastasis [45]. In our study, C5 was more accumulated at the tumor margin. Whether the C5 subset can promote CRC metastasis remains to be explored.

We were also aware of some limitations of this study. Currently, the prognostic data of public datasets were based on bulk RNA sequencing, and lacked prognostic data in scRNA-seq dataset. These limitations result in deficiency of clinical results for cell subsets. Although some algorithms such as CYBERSORTx were developed to impute cell fractions of bulk RNA samples based on scRNA-seq dataset, it still could not make a clear distinction between two cell subsets with similar gene signatures and could lead to inaccurate results. Therefore, the prognostic data for scRNA-seq dataset is in an urgent need and it would help guide treatment decisions.

In conclusion, our study revealed the characteristics of the MPS in CRC and advanced the understanding of colorectal tumor microenvironment. Further study on TIMPs and TF regulons in mononuclear phagocytes is required towards deepening our understanding of MPS.

## Methods

### Ethics statement

All experimental procedures were in accordance with the Declaration of Helsinki. Ethical approval was obtained from the Institutional Review Board of Tongji Medical College, Huazhong University of Science and Technology (permission number: S002).

### Human specimens

All our samples were obtained from the Union Hospital of Tongji Medical College, Huazhong University of Science and Technology, Wuhan, China. Three patients (WHU1, WHU2, WHU3) were pathologically diagnosed CRC and specimens (1*1*1 cm) were took directly from the tumor and collected for scRNA-seq. Clinical information of patients were included

**Table 1. Preoperative and pathologic characteristics of CRC patients.**

| | WHU1 | WHU2 | WHU3 |
|---|---|---|---|
| Gender | female | male | male |
| Age at diagnosis | 70 | 66 | 76 |
| Site of tumor | Ascending colon | rectum | descending colon |
| Time of surgery | 2020/12/14 | 2021/1/18 | 2019/11/26 |
| Pathology | moderately differentiated adenocarcinoma | moderately poorly differentiated adenocarcinoma | moderately differentiated adenocarcinoma |
| Stage | II | III | II |
| T | 3 | 3 | 3 |
| N | 0 | 1 | 0 |
| M | 0 | 0 | 0 |
| preoperative chemotherapy | No | No | No |

in Table 1. Tumor tissues of five CRC patients were collected for immunofluorescence staining.

## Immunofluorescence staining

Immunofluorescence staining of CRC was performed according to the general protocol. After antigen retrieval and autofluorescence quenching, tissue sections were incubated with BSA for 30 min and then incubated with antibody of marker genes of each cluster (Table 2) overnight. Tissue sections were washed with PBS three times and incubated with secondary fluorescent antibodies for 50 min. After washed with PBS three times, nuclei were counterstained with DAPI. The fluorescent photographs were visualized and taken using fluorescence microscope.

## Isolation, culture and differentiation of human blood-derived monocytes

To obtain human blood-derived monocytes, we collected peripheral blood from healthy people. PBMC were isolated through Ficoll density gradient centrifugation (Absin, abs930-200mL). Then monocytes were isolated from PBMC using Human Monocyte Isolation Kit (Stem Cell, Catalog #19359). Monocytes were counted and cultured in RIPM medium with

**Table 2. Antibodies information.**

| Antibodies | SOURCE | IDENTIFIER |
|---|---|---|
| CD14 Monoclonal antibody | proteintech | Cat No. 60253-1-Ig |
| S100A8 Polyclonal antibody | proteintech | Cat No. 15792-1-AP |
| S100A9 Polyclonal antibody | proteintech | Cat No. 26992-1-AP |
| GLUT3 Polyclonal antibody (SLC2A3) | proteintech | Cat No. 20403-1-AP |
| Mitoferrin 1 Polyclonal antibody (SLC25A37) | proteintech | Cat No. 26469-1-AP |
| RETN Polyclonal antibody | proteintech | Cat No. 18170-1-AP |
| PLAC8 Polyclonal antibody | proteintech | Cat No. 12284-1-AP |
| C1qA Polyclonal antibody | proteintech | Cat No. 11602-1-AP |
| C1qC Monoclonal antibody | proteintech | Cat No. 66268-1-Ig |
| APOE Polyclonal antibody | proteintech | Cat No. 18254-1-AP |
| APOC1 Polyclonal antibody | proteintech | Cat No. 16775-1-AP |
| TOP2A Polyclonal antibody | proteintech | Cat No. 24641-1-AP |
| UBE2C Polyclonal antibody | proteintech | Cat No. 12134-2-AP |

**Table 3. Primer sequence of Q- PCR.**

| Primer name | Sequence (5'to3') |
| --- | --- |
| homo TIMP2-F | CTCTGTGACTTCATCGTGCC |
| homo TIMP2-R | TTCTTCTCTGTGACCCAGTCC |
| homo TIMP1-F | CCAGAAGTCAACCAGACCAC |
| homo TIMP1-R | TTCCAGCAATGAGAAACTCCT |
| homo-actB-F | ATCAAGATCATTGCTCCTCCTG |
| homo-actB-R | CTGCTTGCTGATCCACATCTG |

M-CSF (25ng/mL) (Absin, UA040016) for 4 days. Then LPS (10 ng/mL) (Sigma, L5293) and IFN-γ (10 ng/mL) (Absin, UA040053) was added to induce monocytes to differentiate into M1 macrophages. And IL-4 (10 ng/mL) (Absin, UA040026) was added to induce monocytes to differentiate into M2 macrophages. M1 and M2 macrophages were collected after 2 days of culture.

## Quantitative RT-PCR

Total RNA was extracted from cultured cells by the RNA-easy Isolation Reagent (Vazyme, R701-02). Then total RNA was used to synthesize cDNA with the M-MLV (H-) Reverse Transcriptase System (Vazyme, R021-01). The cDNA was subjected to RT-qPCR by the AceQ qPCR SYBR Green Master Mix (Vazyme, Q111-02). The analysis of target gene expression was followed by $2^{-\Delta\Delta CT}$ method. GraphPad Prism 8.3.0 was used for analysis, and all summary data are shown as the mean ± SEM. Student's t tests were assessed to compare the two different groups. The specific primers were listed in Table 3.

## Sorted single-cell suspension preparation

Tumor tissues were immediately cut into approximately 1 mm$^3$ small pieces in RPMI medium after collected from CRC patients. Then each sample was enzymatically digested using Collagenase IV (1 mg/mL), DNase I (100 μg/mL) and hyaluronidase (100 μg/mL) on a 37°C shaker for 1 h. Subsequently, suspension was filtered using a 70 μm cell strainer and centrifuged at 200 g for 5 min. The supernatant was discarded, and then the cells were washed with PBS twice. After the supernatant was removed, the cell pellet was suspended in 3 mL red blood cell lysis buffer and incubated at 4°C for 5 min to lyse red blood cells. With 3 mL RPMI medium added, suspension was centrifuged at 200 g for 5 min. After washing twice with PBS, supernatant was removed and the cell pellet was resuspended in buffer (PBS with 0.5% BSA and 2 mM EDTA).

　　Single-cell suspension was sorted using magnetic bead sorting according to the manufacturer's protocol. Cell count of suspension was counted and then Centrifuge cell suspension at 200 g for 5 min. After removing supernatant completely, cell pellet was resuspended to $10^7$ total cells in 100 μL of buffer and then incubated with biotinylated Anti-Human-CD14 at 4°C for 5 min. Next, suspension was washed with buffer twice and centrifuged at 200 g for 5 min. After supernatant moved, cell pellet was resuspended in 80 μL of buffer per $10^7$ total cells and incubated with Anti-Biotin MicroBeads at 4°C for 15 min. Then suspension was washed with buffer twice and centrifuged at 200 g for 5 min. Cell pellet was resuspend to $10^8$ cells in 500 μL of buffer after supernatant moved. Labeled suspension was applied onto MACS column that rinsing with the appropriate amount of buffer. After unlabeled cells pass through, column was washed with buffer three times and placed on a suitable collection tube. Add the 5 mL buffer onto the column and immediately flush out the magnetically labeled cells by firmly pushing

the plunger into the column. Finally, the sorted single-cell suspension was counted and diluted in an appropriate concentration.

## Single-cell RNA sequencing

scRNA-seq was performed according to the manufacturer's protocol. Single-cell suspensions were mixed with Chromium Single cell 3′ Reagent v3 kits and then were loaded on Chromium Controller to generate single-cell gel beads in emulsions (GEMs). Cell lysis and reverse transcription reactions were performed in GEMs to generate barcoded full-length cDNA. Then GEMs were disrupted and cDNA was amplified. The amplified cDNA was fragmented, end-repaired, A-tailed, and ligated to an index adaptor to generate barcoded scRNA-seq libraries. Subsequently, scRNA-seq libraries were sequenced on the Illumina NovaSeq platform and raw sequencing data were generated for downstream analyses.

## Statistics

**scRNA-seq data processing and quality control.** Raw data was processed using Cell Ranger toolkit. With cellranger count function, raw data was mapped to the reference genome GRCh38 and generated the gene-cell UMI matrix for downstream analyses.

The UMI matrix was converted into a Seurat object using the R package Seurat2 (version 4.1.0) [46]. To filter out possible empty droplets, low-quality cells, and possible multiplets, cells with <200 or >6000 features and with <500 UMI counts were excluded. Also, to filter out cells of low quality, cells with >20% of their transcripts coming from mitochondrial genes were excluded from analysis. Cell-type annotation was performed based on R package SingleR (version 1.8.1) [12]. Monocytes and macrophages were selected as the research object. Finally, 5153 cells were remained for downstream analyses.

As the dissociation and digestion during single-cell suspension preparation would lead to high levels of immediate early genes (Fos, Jun and other activating protein 1 complex genes) and heat-shock proteins (HSPs) [47], these dissociation and stress associated genes were excluded from further analysis [48].

To remove potential batch effect, data from three patients were integrated using a Seurat approach. The UMI matrix was normalized and separated by patient sample. With FindIntegrationAnchors function of R package Seurat, Top 2000 high variable genes were used to create anchors. Then, we use IntegrateData function to these matrixes were integrated into a new matrix without batch effect.

The principal component analysis (PCA) was performed to reduce dimensionality of the integrated matrix. Top 30 principal components determined by ElbowPlot function were used in the downstream analyses. With resolution = 0.2 and Louvain clustering, six clusters were identified using FindClusters function. And then clustering results were visualized with two-dimensional UMAP plots.

**Bioinformatics analysis of single-cell sequence data.** Our single-cell sequence data, public scRNA-seq datasets of blood from healthy people (GSE167363), colon from healthy people (GSE214695), blood and normal colon from CRC patients (GSE161277), CRC samples (GSE132465), CRC patients treated with or without PC (GSE178318), CRC patients before and after chemotherapy (GSE232525), BLCA patients (GSE190888), GBM patients (GSE162631), HNSC patients (GSE234933), KIRC patients (GSE121636), LIHC patients (GSE242889), LUSC patients (GSE123904), PAAD patients (GSE214295), STAD patients (GSE234129) and spatial transcriptomics data from GSE226997 in Gene Expression Omnibus (GEO) were used for bioinformatics Analysis.

**Functional enrichment analysis.** We performed functional enrichment analysis to investigate biological characteristic of different clusters. Gene Set Variation Analysis (GSVA) of biological process GO term was performed using R package GSVA (version 1.42.0).

**Polarization score of MPS.** To evaluate M1/M2 polarization ability of mononuclear phagocytes. We applied GSVA to calculate M1/M2 polarization score. The reference gene set of M1/M2 polarization was described by Sun et al [49] and all the parameters were set as default.

**Trajectory analysis.** Pseudotime analysis and RNA velocity analysis were used to infer developmental trajectory of mononuclear phagocytes. Pseudotime analysis was performed based on R package Monocle2 (version 2.22.0) [50]. The UMI matrix was converted into a CellDataSet object and genes with mean expression < 0.1 were filtered out. Then significantly changed genes identified by the differentialGeneTest function were used to order mononuclear phagocytes in pseudotime analysis. Cell trajectories were constructed and visualized with plot_cell_trajectory function. Key genes during the differentiation progress of mononuclear phagocytes were detected with Monocle2 plot_pseudotime_heatmap function. RNA velocity analysis was performed based on python package velocyto (version 0.17) and python package scvelo (version 0.2.5) [51]. In this process, spliced reads and unspliced reads of single-cell sequence data were calculated based on aligned bam files. Then RNA velocity values were estimated with dynamical model and the velocities were visualized in two-dimensional plot.

**Regulatory network analysis.** To explore gene regulatory network of mononuclear phagocytes, we performed single-cell gene regulatory network analysis using R package SCENIC (version 1.3.0) [26]. Following SCENIC tutorial, candidate regulatory modules were inferred from co-expression patterns between genes, and co-expression regulatory modules were refined using TF regulons motif information. Then, AUC score of each TF regulons in each cell was calculated with AUCell function.

**Metabolic analysis.** To explore metabolism activity of clusters, we applied scMetabolism (version 0.2.1), a new algorithm used to quantify metabolism activity at the single-cell resolution [27], on single-cell sequence data. Metabolism score of each cell was calculated with sc. metabolism.Seurat function.

**Correlation analysis.** Ten major cell types of TCGA CRC cohorts were imputed with CYBERSORTx, then Pearson correlation analysis was performed to assess the relationship among the proportions of ten cell types.

**Survival analysis.** Transcriptome data and clinical data from TCGA were used to evaluate the prognostic performance of genes and cell clusters. The TCGA data were obtained from UCSC Xena (http://xena.ucsc.edu/). Transcriptome data was normalized to TPM data. The optimal cutpoint was determined by R package survminer (version 0.4.9). Then samples were grouped into high expression and low expression groups according to the cut-off expression of the gene. Kaplan-Meier survival curves was used to evaluate the correlation between genes and survival.

To evaluate the role of cell clusters in predicting the prognosis of cancers, we apply CYBERSORTx to impute abundance of cell clusters [28]. Then samples were grouped into high abundance and low abundance groups based on cut off value determined by abundance. Kaplan-Meier survival curves was also used to evaluate the correlation between abundance and survival.

## Supporting information

**S1 Fig. Six clusters of MPS are Identified. (**A). Quality control of single-cell RNA sequence data. **(**B). UMAP plots of mononuclear phagocytes before integration (left) and after

integration (right). Cells are colored by samples. (C). UMAP plots showed expression levels of marker genes of mononuclear phagocytes (CD14), intestinal dendritic cells (CX3CR1) and neutrophils (MPO and CEACAM8, also named CD66b).
(TIF)

**S2 Fig. Marker genes of mononuclear phagocytes subsets are verified in CRC tissues.** (A). UMAP plots showed expression levels of marker genes of each cluster. (B). Immunofluorescence staining of CD14 and marker genes of each cluster in CRC tissues confirmed the existence of six clusters. White arrows point cells of each cluster. Scale bars, 20 μm.
(TIF)

**S3 Fig. M1-like and M2-like cell of MPS is verified in public scRNA-seq dataset and human peripheral blood-derived macrophages.** (A). UMAP plot of mononuclear phagocytes in GSE132465 CRC patients. Cells are colored by clusters. (B). Scatterplot showed M1 polarization and M2 polarization score of each cluster in GSE132465 CRC patients. The mean score of each cluster is represented by a color-coded triangle. (C). UMAP plot of mononuclear phagocytes in GSE132465 CRC patients, color-coded by M1-like and M2-like cell. (D). Heatmap displaying differential expression pattern of genes in M1-like and M2-like cells. (E). UMAP plots showed expression levels of TIMP1 and TIMP2 in GSE132465 CRC patients. (F). Expression levels of TIMP1 and TIMP2 in human peripheral blood-derived M1 and M2 cells (N = 3), *P < 0.05. The statistical significance was obtained by Student's t-tests.
(TIF)

**S4 Fig. Spatial distribution of MPS in CRC tissue.** (A). Spatial plots of another CRC sample showed areas of tumor and stroma based on tumor score in tissue sections (top), and distribution signature of TIMP1 and TIMP2 in CD14-expressed area (bottom). (B). Violin plots showed expression level of TIMP1 and TIMP2 in different CD14-expressed areas. (C). Spatial feature plots showed distribution signature of six clusters in tissue sections, respectively. ns represent no significance, * represent p value < 0.05, ** represent p value < 0.01, *** represent p value < 0.001.
(TIF)

**S5 Fig. Dynamic changes of clusters and genes during the differentiation of MPS.** (A). Split pseudotime plot of mononuclear phagocytes showed stages of each cluster. (B). Scatterplots showed downregulated inflammatory genes and upregulated immunosuppression-related genes in the differentiation of mononuclear phagocytes. Cells are colored by clusters.
(TIF)

**S6 Fig. Similar clusters are identified in MPS of public scRNA-seq dataset.** (A). UMAP plot of mononuclear phagocytes in GSE178318 CRC patients. Cells are colored by clusters. (B). Heatmap showed similar expression characteristics of marker genes between GSE178318 and our data. (C). UMAP plot of mononuclear phagocytes in GSE232525 CRC patients. Cells are colored by clusters. (D). Heatmap showed similar expression characteristics of marker genes between GSE232525 and our data.
(TIF)

**S7 Fig. C5 is a unique cluster different from SPP+ macrophages.** (A). UMAP plot showed expression level of SPP1 and RNASE1 in MPS. (B). Split UMAP plots showed expression levels of SPP1 in each cluster.
(TIF)

**S8 Fig. Identification of mononuclear phagocytes subsets in eight types of tumor.** (A-H). UMAP plot (left) and dot plot (right) showed mononuclear phagocytes subsets in BLCA (A), GBM (B), HNSC (C), KIRC (D), LIHC (E), LUAD (F), PAAD (G) and STAD (H). (TIF)

## Author Contributions

**Conceptualization:** Tiantian Ji, Haoyu Fu, Zheng Wang.

**Data curation:** Tiantian Ji, Haoyu Fu, Liping Wang, Jinyun Chen.

**Formal analysis:** Tiantian Ji, Haoyu Fu, Shaobo Tian.

**Funding acquisition:** Tiantian Ji, Guobin Wang, Lin Wang, Zheng Wang.

**Investigation:** Tiantian Ji, Haoyu Fu.

**Methodology:** Tiantian Ji, Haoyu Fu.

**Project administration:** Lin Wang, Zheng Wang.

**Resources:** Lin Wang, Zheng Wang.

**Software:** Haoyu Fu, Liping Wang.

**Supervision:** Lin Wang, Zheng Wang.

**Validation:** Tiantian Ji, Haoyu Fu, Zheng Wang.

**Visualization:** Haoyu Fu.

**Writing – original draft:** Tiantian Ji, Haoyu Fu.

**Writing – review & editing:** Guobin Wang, Lin Wang, Zheng Wang.

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
