## [Decision Letter · Decision Letter 0]

1 Nov 2023

Dear Dr Wang,

Thank you very much for submitting your Research Article entitled 'Single-cell RNA profiling reveals classification and characteristics of mononuclear phagocytes in colorectal cancer' to PLOS Genetics.

The manuscript was fully evaluated at the editorial level and by independent peer reviewers. The reviewers appreciated the attention to an important problem, but raised some substantial concerns about the current manuscript. Based on the reviews, we will not be able to accept this version of the manuscript, but we would be willing to review a much-revised version. We cannot, of course, promise publication at that time.

If you decide to revise the manuscript for further consideration at PLOS Genetics, please aim to resubmit within the next 60 days, unless it will take extra time to address the concerns of the reviewers, in which case we would appreciate an expected resubmission date by email to plosgenetics@plos.org.

We are sorry that we cannot be more positive about your manuscript at this stage. Please do not hesitate to contact us if you have any concerns or questions.

Yours sincerely,

David J. Kwiatkowski

Section Editor

PLOS Genetics

David Kwiatkowski

Section Editor

PLOS Genetics

Reviewer's Responses to Questions

**Comments to the Authors:**

Reviewer #1: Thank you for the opportunity to review “Single-cell RNA profiling reveals classification and characteristics of mononuclear phagocytes in colorectal cancer”. The study builds on current understanding in the field that tumor-associated macrophages in colorectal cancer have properties that can drive tumorigenesis. The focus of the current study examines gene expression of CD14-enriched populations from colon tumors excised from three patients. Six clusters are identified and are suggested to arise from developmental progression in order from 1-6. One cluster, C5, is selected for further analysis and two genes enriched in these cells (APOE and RNASE1) are shown to be linked to poor prognosis in CRC. Additionally, using software that infers the abundance of specific cell types among datasets of publicly available bulk-RNAseq, the increased abundance of the C5 cluster is inferred to be associated with poor survival probably in other cancers. While there are some suggestive findings in the present study, the conclusions are not well supported by the data shown. It is also unclear that the findings significantly build upon current understanding in the field. Moreover, there are a number of concerns that bring into question the identities of the cell types examined.

Major concerns are as follows:

The populations studied are enriched by CD14-bead selection, but the data are not examined for the inclusion of CD14-low populations, which would include neutrophils. Neutrophils express the S100 family of proteins and the possibility that neutrophils are present among the clusters examined is not excluded. Also, in some cases, CD14 is expressed by dendritic cells and the dataset is not evaluated to ensure that dendritic cells are excluded. Dendritic cells are a constituent of the mononuclear phagocyte population and the data should be explicit about whether or not these are present—as written, the study would suggest that only macrophages and monocytes are components of the mononuclear phagocyte network, which is incorrect. An evaluation of CSF1R is performed, and it is proposed that these are exclusively expressed by only some clusters. This would seem in contrast with current literature indicating that mononuclear phagocytes all express CSF1R to some respect, and can be further upregulated in pathologic settings (Combes, Immunother Adv, 2021). The authors should clarify the criteria set for CSF1R positive and negative populations. The negative populations may not belong to the mononuclear phagocyte network.

Another major concern is the absence of an attempt to align the clusters found in the current study with mononuclear phagocytes (including those which are CD14+) known to reside in the healthy intestine. These would include analyses to designate populations distinguished by standard criteria of expression of CD16, HLA-DR, CD11b, CD88, CD64, CD1c and other standard criteria used to subset intestine-resident mononuclear phagocytes (described in many publications, including Bujko, J exp med, 2018; Garrido-Trigo, Nat Comm, 2023, Hegarty, Nat Review Gastro and Hepato, 2023). This will fortify the data in specifying which populations are monocytes, transitioning monocytes and other developmental fates as the authors suggest. Whether or not these standard criteria can be applied to the dataset, there is still a need to understand whether the clusters described in the current report exist in the healthy intestine, which can be ascertained from analysis of publicly available datasets.

Currently, the authors do not fully describe the clusters. Gene expression the authors put forward to distinguish clusters is not rigorously tested and/or attempted to align with published literature on those genes, as they are expressed by macrophages. Given that there is existing literature on several of the cluster-defining genes reported here, the authors need to give more attention to previous reports to expand on current understanding. The authors should attempt to stain for more cluster-defining genes in tissue samples. The authors should also investigate whether gene expression of other similar genes align with their findings, such as the author’s claims for higher phagocytic potential. Information on other gene expression for complement genes would also be of interest to the field. To further address the study’s focus on macrophage heterogeneity, the authors should align and discuss their findings as compared to the published literature on M2-like clusters C4 to C6 with the M2a, M2b, M2c, and M2d stratificaitons previously reported. An effort to align results with that which is already known will advance the field in regards to better understanding of TAMs subsets.

There is a need to describe the patients and how these differences might align with differences in the cluster abundance. For example, C5 was the dominant TAMs in WUH1, while WUH2 and WUH3 patients have more abundant M1-like macrophage clusters. Could the author please comment whether the tumor stages or progression of those three patients as they potentially affect the TAMs population distribution?

The authors refer to the C5 cluster as immunosuppressive, but provide no functional data to support that conclusion. Since C4 and C6 also shows M2-like macrophage phenotype, it will be important for the authors to show whether C4 or C6 cluster (and all the other clusters) correlated with poor prognosis as well. This is particularly important because C4 shares a lot of features with C5, including the TF regulons reported identified by the authors. Moreover, the author’s focus on C5 is unclear, given that the data shown suggests the C4 population is both more abundant and responsive to PC treatment (Fig 5).

Fig 5: The authors analyzed scRNAseq dataset from preoperative chemotherapy (PC) treated patients and showed C5 was significantly reduced with tumor shrinkage. The caveat that PC has direct effect to diminish population abundance of C5 is not addressed. Can the authors provide further validation of their conclusions by analyzing other publicly available datasets from CRC patients receiving other methods of treatment?

Fig 2f: Conclusions of the imaging data are unclear, given that there is limited overlap of stains with CD14. There is a need for quantification to support the author’s conclusions, as well as additional samples tested.

The author’s suggestion that the C6 cluster is proliferative may indicate the need to perform a second pseudotime analysis without this population being considered. Proliferation of mature macrophages may indeed be altered in the tumor environment (Soncin, Nature Communications, 2018), and the author’s study is an opportunity to further test this hypothesis. Another hypothesis to be tested would be that the proliferating populations are not dependent on monocyte progenitors and thus, inappropriately placed on the pseudotime analysis. These hypotheses should be tested in this report.

M1 and M2 macrophages are both terminally differentiated subsets. Although the authors found different expression pattern of TIMP1 and TIMP2 in M1-like and M2-like macrophages, the analysis in figure 2 did not support the idea that TIMP1 and TIMP2 are involved in mononuclear phagocytes differentiation or M1 to M2 transition plasticity. Especially, by the trajectory and RNA velocity analysis, the authors claimed a development pathway from C1 to C5, however, the expression pattern of TIMP1 and TIMP2 was varied among clusters, which did not support the trajectory. The authors should consider the use of human monocyte-derived cultures to test their hypotheses.

The authors group CD86 as a immunosuppressive gene, but this is not the typical classification for the CD86 T cell co-stimulatory pathway, and the citation for the author’s functional annotation for CD86 as immunosuppressive in colorectal cancer is not described. By contrast, while CD274’s role in T cell exhaustion is well known, CD274’s role in colorectal cancer is not well established. Since CD274 is naturally expressed by intestinal macrophages in conditions of health, the authors should provide citations to expand on the understanding of the roles of CD274 in colorectal cancer as they are interpreting it in this study.

Fig 6: The author’s comparisons of C5 versus non-C5 is inappropriate, given that the non-C5 are heterogenous. Comparisons to each individual cluster would be more informative for the proposed differences. Also, GO terms should be provided for all the clusters to best understand their distinctions.

Fig S5. The author’s conclusions that the C5 cluster is distinct from SPP1+ macrophages is unclear, given that the data shows clear overlap between SPP1 expression and the C5 cluster. There appears to be some alignment, which can be discussed in the results appropriately.

Fig7: The authors need to clarify the criteria used to establish “mononuclear phagocytes” and “non mononuclear phagocyte” for the APOE and RNASE1 analysis. The authors should also specify whether the “non mononuclear phagocytes” are immune cells, which would be the appropriate population comparison. A comparison of the tumor expression would also be informative.

Last, the APOE and RNASE1 expression in CRC described here is interesting. For their conclusions beyond CRC, the data are suggestive but far from conclusive. While the existence of a “universal” tumor-associated macrophage associated with poor prognosis in solid tumors would be of interest, there is not enough data to support the authors claims. Intestine is known to have unique populations of mononuclear phagocytes, and it cannot be assumed that these are present (or of similar identity) in non-intestinal tissues. Moreover, gene expression that distinguishes the different populations of mononuclear phagocytes is well known to be tissue specific and assumptions of identity (without further validation) should be minimal. There needs to be transparent criteria of matching for the C5 cluster and transparency of criteria sorting into high vs. low abundance of the C5 cluster, and whether it is found in all datasets evaluated, to determine whether the cluster is of relevant amounts to be considered physiologically relevant in non-intestinal tissues.

In line 113, the author claimed that in WUH3 patient had a higher portion of C2 but by figure 1E, the most abundant population in WUH3 was C3.

Reviewer #2: I have read with interest this study that concerns the classifications of different populations of TAMS in colorectal cancer. By using the scRNAseq technology, the authors reported 6 different TAMs subsets in colorectal cancer showing, in particular, that there was an immunosuppressive subset associated with worse survival.

I believe that this manuscript is overall well-written, and it might be of interest to the readership of PLOS Genetics. However, I would suggest the following changes:

1. The clinical data of the three CRC patients should be added. General clinical, pathological, and surgical data should be given.

2. Which were the TNM stages?

3. Were these 3 patients homogenous? Had they similar clinical and pathological characteristics?

4. Did they get chemotherapy before colectomy?

5. The protocol of specimen retrieval should be given. From which part of the colon were the specimen taken? Directly from the tumor o from the peri-tumoral areas?

6. The following papers should be considered and eventually added / commented in the discussion: 1) Cortese N, et al. Cancer Immunol Res. 2023 Apr 3;11(4):405-420; 2) Donadon M, et al. J Exp Med. 2020 Nov 2;217(11):e20191847; 3) Murray PJ. J Exp Med. 2020 Nov 2;217(11):e20201259)

7. The authors should investigate more the relationship between TIMPs, the C5 subset and the associated lipid metabolism. Since they reported such relationships, a lipidomic approach should be given otherwise it remains a speculation.

8. What is really missing in this paper, and it would be nice to have, is a kind of validation of authors’ scRNAseq data. FACS, PCR and/or IHC analyses would be of added value.

Reviewer #3: This is an interesting article that adds to the growing body of knowledge with regards to macrophage/TME biology in the context of solid tumors. Like other papers, they describe and annotate macrophage populations within CRCs using scRNA-Seq. However, the authors make some major claims about the significance of their data which need to be backed up with deeper analysis.

The manuscript could be improved by clarifying the following:

1. It appears that the clustering results are highly dependent on resolution parameters in Seurat -- the authors observed clusters C1-C6 but it appears that C1-3 are pretty similar to each other (vs C4-5, and C6). Have the authors assessed the robustness of their clustering? Was Leiden or Louvain clustering used to generate them?

2. Fig 2A and 2B appear to show opposing results: 2A -- where the triangle points show the mean polarization of each cluster, do not appear to have any significant polarization in M1 and have some difference in M2 polarization. Yet, in 2B, it is clear that M1 and M2-classified cells clearly separate in UMAP space. From Fig 1D it looks like C1-3 are M1 and the rest are M2 but 2A does not reflect that at all. Perhaps the authors can clarify.

3. There seems to be a typo on line 134: "special" should read "spatial". I'd advise the authors to search for this elsewhere in the manuscript.

4. It's not clear to me why the authors used spatial analysis strictly for M1/M2 cells, when it appears that the main novelty of the manuscript is the discovery of additional macrophage clusters C1-C6. The section "Tissue inhibitor of metalloproteinases (TIMPs) participate in the differentiation of M1-like and M2-like mononuclear phagocytes" appears to attempt an alignment between C1-6 clusters with canonical M1/M2 polarization, but then does not follow up with how their new findings are interpreted with the spatial information. How are C1-6 spatially distributed? Considering that marker genes were found in each C1-6 cluster why was spatial analysis not done? The authors performed IF in Fig 5C so they have the capability.

5. The authors should perform KM-curve analysis using marker genes of their discovered C1-6 clusters, otherwise it's not convincing that C5 itself has a worse prognosis. Otherwise, you could argue that any markers associated with macrophages lead to a worse prognosis.

**Have all data underlying the figures and results presented in the manuscript been provided?**

Reviewer #1: Yes

Reviewer #2: Yes

Reviewer #3: Yes

PLOS authors have the option to publish the peer review history of their article (what does this mean?). If published, this will include your full peer review and any attached files.

Reviewer #1: No

Reviewer #2: No

Reviewer #3: No

---

## [Decision Letter · Decision Letter 1]

8 Feb 2024

Dear Dr Wang,

We are pleased to inform you that your manuscript entitled "Single-cell RNA profiling reveals classification and characteristics of mononuclear phagocytes in colorectal cancer" has been editorially accepted for publication in PLOS Genetics. Congratulations!

Yours sincerely,

David J. Kwiatkowski

Section Editor

PLOS Genetics

David Kwiatkowski

Section Editor

PLOS Genetics

Comments from the reviewers (if applicable):

Reviewer's Responses to Questions

**Comments to the Authors:**

Reviewer #2: I have read the R1 version of the paper and the point-by-point authors' reply. I thank the authors for providing all the information requested, which enforced the paper that now appears to be more clear and convincing.

I have no more comments for this round.

Reviewer #3: The authors have sufficiently answered my questions. Thank you for the prompt response!

**Have all data underlying the figures and results presented in the manuscript been provided?**

Reviewer #2: Yes

Reviewer #3: Yes

PLOS authors have the option to publish the peer review history of their article (what does this mean?). If published, this will include your full peer review and any attached files.

Reviewer #2: No

Reviewer #3: No

**Data Deposition**

http://datadryad.org/submit?journalID=pgenetics&manu=PGENETICS-D-23-00566R1

**Press Queries**

---

## [Editor Report · Acceptance letter]

21 Feb 2024

PGENETICS-D-23-00566R1 

Single-cell RNA profiling reveals classification and characteristics of mononuclear phagocytes in colorectal cancer 

Dear Dr Wang, 

We are pleased to inform you that your manuscript entitled "Single-cell RNA profiling reveals classification and characteristics of mononuclear phagocytes in colorectal cancer" has been formally accepted for publication in PLOS Genetics! Your manuscript is now with our production department and you will be notified of the publication date in due course.

With kind regards,

Zsofi Zombor

PLOS Genetics

On behalf of:
